# Finite Element Modeling of Residual Hearing after Cochlear Implant Surgery in Chinchillas

**DOI:** 10.3390/bioengineering10050539

**Published:** 2023-04-27

**Authors:** Nicholas Castle, Junfeng Liang, Matthew Smith, Brett Petersen, Cayman Matson, Tara Eldridge, Ke Zhang, Chung-Hao Lee, Yingtao Liu, Chenkai Dai

**Affiliations:** Department of Aerospace and Mechanical Engineering, University of Oklahoma, Norman, OK 73019, USA; ncastle@ou.edu (N.C.);

**Keywords:** inner ear, cochlear implant, chinchilla, finite element, insertion angle

## Abstract

Cochlear implant (CI) surgery is one of the most utilized treatments for severe hearing loss. However, the effects of a successful scala tympani insertion on the mechanics of hearing are not yet fully understood. This paper presents a finite element (FE) model of the chinchilla inner ear for studying the interrelationship between the mechanical function and the insertion angle of a CI electrode. This FE model includes a three-chambered cochlea and full vestibular system, accomplished using µ-MRI and µ-CT scanning technologies. This model’s first application found minimal loss of residual hearing due to insertion angle after CI surgery, and this indicates that it is a reliable and helpful tool for future applications in CI design, surgical planning, and stimuli setup.

## 1. Introduction

### 1.1. Cochlear Electrodes Importance and Trauma

More than 5% of the world’s population suffers from disabling hearing loss, amounting to over 430 million people [1]. In cases where hearing aids are no longer useful or sufficient, cochlear implant (CI) surgery is the standard procedure for the treatment of severe hearing loss. Modern CI surgery often significantly improves patients’ health-associated quality of life [2,3,4,5,6]. However, it is also known that CI surgery can cause varying levels of trauma or cochlear obstruction that affect the residual mechanical function of the inner ear [7]. The magnitude of this disruption could be partially dependent on the insertion depth of the implant [5,8].

### 1.2. Effect of CI Surgery on Residual Hearing

Most sources report that the magnitude of CI surgery’s effect on residual hearing is largely dependent upon whether cochlear trauma takes place during the CI surgery. Trauma is usually attributed to the dislocation of the CI electrode from the scala media or vestibuli [9]. Dislocation can arise due to a variety of factors thus necessitating the correct choice of an electrode, the surgical technique, and the insertion angle [10]. An important consideration is the morphology of the patient’s cochlea, as shorter and smaller cochleae tend to have higher rates of intracochlear dislocation when fully inserted [11]. In most modern cases, CI electrodes are correctly placed in the scala tympani with minimal trauma. However, a systematic understanding of the effects of typical CI electrode placement on the finer sensitivity of the basilar membrane could be an important step toward further improvement of CI electrode design.

### 1.3. Effect of Insertion Angle on CI Effectiveness and Residual Hearing

Insertion angle is a major contributor to CI effectiveness [7]. When longer electrode models are selected, typically with an angle of insertion greater than 540 degrees, lower frequencies become more perceptible to patients, music becomes more enjoyable, and the quality of life increases compared to those patients with shorter CI electrodes [12]. In cases where CI surgery results in minimal-to-no trauma, and the patient is healthy with few underlying conditions, very few side effects are reported other than postoperative vertigo and nausea [13].

### 1.4. Advantages of the FE Method over In Vivo Testing

The use of laboratory animals is a necessary part of medical research, as it enables scientists to explore new treatments before progressing to human trials. However, animal testing is a complex issue that raises many ethical and logistical concerns, particularly regarding the welfare and cost of laboratory animals. Animal testing must be carried out in accordance with strict ethical guidelines to ensure that any suffering is minimized [14]. In medical research, it is often necessary to purchase many expensive research-grade animals, making their use particularly expensive, especially when considering the long-term management of an animal facility [15]. Furthermore, the quality of laboratory animals can be compromised by unethical practices, making studies less productive and reproducible. Finite element modeling is a solution to both the monetary and ethical problems involved with animal research [16,17]. FE modeling is cost-effective, ethical, reproducible, and safe. Models can be precisely manipulated at will in a relatively short time frame to account for a variety of different variables and conditions, some of which may not be foreseen prior to beginning the modeling. Simulations can be run as many times as researchers desire with little-to-no variation in the model’s geometry between iterations, something impossible when using multiple animals in a study [18]. In animal testing, this kind of iterative process can also be quite expensive, involving the purchase of many animals. The FE method can be applied without any harm to the animal subjects, as the medical imaging of delicate structures can be obtained non-invasively. Imaging can be shared among institutions, further reducing the number of animals needed for FE modeling. While not a replacement for animal testing, it is clear that, in early stages of research, the FE method should be explored prior to in vivo testing on animal or human subjects.

### 1.5. Prior FE Models

Over the past decade, substantial research progress has been made to advance inner ear computational modeling. Specifically, finite element (FE) modeling allows the intricacies of the inner ear’s mechanics to be reduced to simpler phenomena that can be verified with clinical results. One such model found that material, geometric design, insertion speed, and friction coefficients were the greatest factors influencing residual hearing preservation [19]. A previously developed finite element model that focused on residual hearing found that cochlear implants most dramatically affect the residual hearing at extreme frequencies of human hearing [20]. This model provided a good first step towards further FE analysis of residual hearing after CI surgery, although it did not necessarily agree with the results of other models where residual hearing was found to be less affected by the simple presence of a CI electrode and more affected by the trauma caused during insertion [19,21]. However, the previous models did not examine the effect of varying cochlear electrode insertion angles between patients. These unexplored results could provide important metrics for clinical use. Therefore, a comprehensive finite element model capable of simulating hearing function with a variety of insertion angles is an essential step in the improvement of cochlear implant design and surgery.

### 1.6. Chinchilla as an Animal Model

The chinchilla is used as the animal model in this study. The chinchilla is commonly used as an analog for a human in hearing and balance studies due to its similar number of turns in the cochlea, structure of semicircular canals, singular primary crista, and hearing range [22,23,24]. Chinchillas are commonly used as an analog in FE analysis due to their large bulla and easy availability, which allow for very fine resolution of models if scanned with a µ-MRI or µ-CT machine. For the preliminary design of the electrode and efficacy measurement, the chinchilla is a good animal model on account of its similarity to the human structure, hearing frequency range, low cost, and larger supply source (compared with primates). Furthermore, a chinchilla computational model will be useful to conduct virtual experiments and eventually reduce the extensive use of chinchillas.

### 1.7. Focus of the Study

This study focuses on the effect of cochlear electrode insertion depth on the residual mechanical function of the cochlea in a chinchilla computational model. The unimplanted model is demonstrated here first and compared to the expected response curves to demonstrate its initial validity. The analysis then focuses on the effects of cochlear implantation on residual hearing.

## 2. Materials and Methods

### 2.1. Data Source and Segmentation

The model geometry was generated through 3D reconstruction of a single, adult chinchilla. The CT scans were acquired at 12 µm voxel size, and the µ-MRI scans were acquired at 30 µm voxel size. Achieving this voxel size with an adequate reduction in feedback for the µ-MRI required 26 hr of acquisition in an 11.7 Tesla magnet SIEMENS MRI machine located in Avanto, Munich, Germany. The µ-MRI and µ-CT scans of the chinchilla bulla were segmented using a program called 3D Slicer [25]. The images were separated into segments representing lymphatic fluid, bone, and nervous tissue. Sample-segmented µ-MRI and µ-CT images are shown in Figure 1 and Figure 2.

**Figure 1 bioengineering-10-00539-f001:**
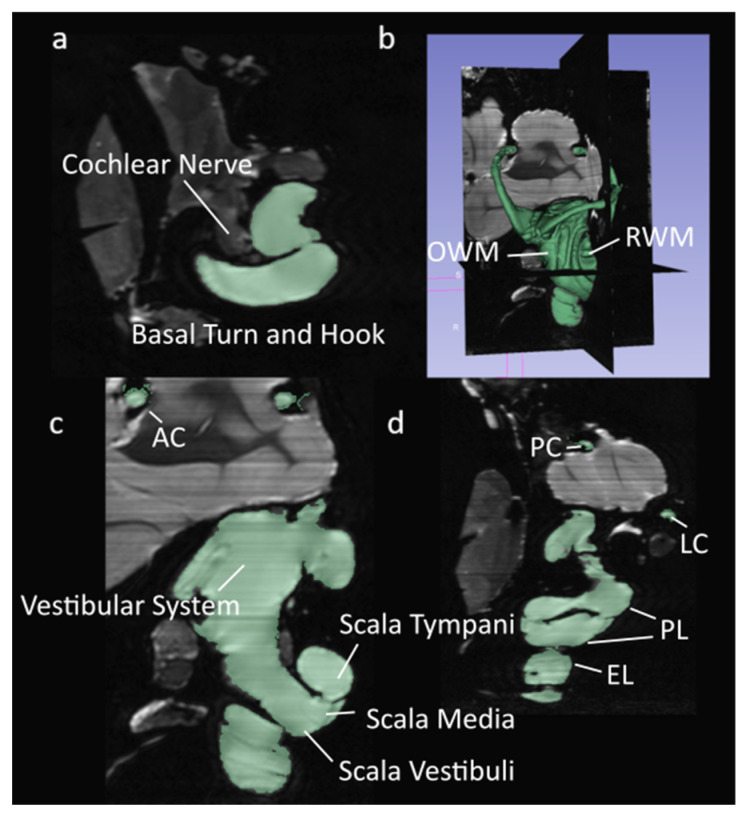
Segmented µMRI scans of the chinchilla subject with key structures labeled. The lymphatic fluid of the inner ear is shown in green. (**a**) Transverse plane; (**b**) 3D view of the entire segmentation; (**c**) Saggital plane; (**d**) Coronal plane. Refer to Table 1 for symbol definitions.

### 2.2. Geometry

The geometry obtained from 3D Slicer was imported into MeshMixer for the smoothing of surfaces. Numerous small bumps and cavities, due either to bone porosity or imaging artifacts, were removed from the boundary of the bony labyrinth. The holes and gaps in the semicircular canals and cochlea were repaired manually to ensure a continuous volume. The curvatures of the repaired sections were matched to those of the surrounding surfaces using tools in the MeshMixer program. For this study, all geometry outside the otic capsule was excluded.

The membranous labyrinth of the semicircular canals was modeled in MeshMixer by creating a copy of the bony labyrinth shrunk by a fraction to create two volumes, one enclosed within the other. The utricle’s shape was modified to maintain proper connectivity with the semicircular canals and the ampullae. The utricle was scaled to accommodate a macula consistent with the descriptions in the literature [26,27]. The saccule was modeled by cross-referencing measurements obtained for humans with data obtained on the saccular macula in the chinchilla [26,28,29]. The membranous labyrinth in the semicircular canals was scaled to ensure a realistic ratio of endolymphatic fluid to perilymphatic fluid by volume. The cupula structures follow the diaphragmatic model and span the entire width and height of the ampullae. The diaphragmatic model is commonly used in the modeling of vestibular mechanics and yields results that closely mirror reality [30,31,32,33]. The maculae were modeled with two distinct layers, a gel layer and an otoconial layer, as was performed previously in a computational model that isolated the maculae for analysis [34]. The reuniting duct was modeled according to measurements found in the literature [35,36,37]. The coordinate system for all figures in this study is described in Figure 3. An annotated model of the completed vestibular system is illustrated in Figure 4.

The cochlea was modeled based primarily on information obtained from µ-MRI imaging. The characteristic ridges on the surface of the bony labyrinth were used to determine the attachment points of the Reissner’s and basilar membranes. The osseous spiral lamina was also clearly defined and marked the inner attachment of both the Reissner’s and basilar membranes. These curves were connected by planes forming a wedge whose superior face represents the Reissner’s membrane and whose inferior face represents the basilar membrane. This shape was compared with that reported in the literature and was confirmed to have the correct structure [38]. The completed model of the cochlea is displayed in Figure 5.

The cochlea was scaled until the basilar membrane was the average length in chinchillas of 18.3 mm along its midline [39]. The thickness of the basilar membrane was varied from 16.5 μm at the base to 5 μm at the tip according to the values given for the pars pectinate in Cochlear Anatomy and Central Auditory Pathways [40].

The dimensions of a MED-EL FLEXSOFT electrode array were scaled to create an analogous implant, which was placed in accordance with an ideal round window insertion in the scala tympani of the cochlea. The cochlear electrode extends almost the full length of the scala tympani with a maximum insertion angle of 900 degrees. This implant was split into 180-degree sections to allow for analysis with varying angles of insertion. Cross-sections at the proximal and terminal ends of the cochlear electrode are shown in Figure 6.

### 2.3. Meshing

Only the components relevant to the mechanical model were meshed. This includes the oval window membrane, round window membrane, cupulas, vestibular maculae, basilar membrane, Reissner’s membrane, utricle, saccule, semicircular canals, cochlea, and cochlear implant electrode array. The mechanical model was meshed with a total of 414,629 tetrahedral elements and 90,696 nodes using the software HyperMesh 2017. Mesh size convergence analysis was not conducted due to the already very fine average element size of approximately 0.2 mm. This size is sufficient when considering the larger element sizes utilized by previous models [41,42]. However, applying mesh size convergence analysis to future iterations of the model may reduce the processing power and time necessary for simulation [43]. The tissues and the electrode array were modeled using the Ansys Solid185 element type, while the fluids were modeled using the Ansys Fluid30 element type. All elements were tetrahedral. Both SOLID185 and FLUID30 tetrahedral elements have 8 nodes, each with 3 degrees of freedom. SOLID185 is a very commonly applied element type in ANSYS, often in structural analysis and in fluid–structure interactions. FLUID30 is accepted as a standard element type for fluids in simulation of fluid–structure interactions. A fine-ruled mesh composed of 8738 elements was chosen for the basilar membrane. All components were assigned their respective thicknesses, meshed, and given proper connectivity using HyperMesh.

### 2.4. Material Properties

Due to the relative scarcity of published data on the material properties of chinchilla inner ear soft tissues, the material properties measured in humans were substituted as needed. All the material properties are shown in Table 2. The mechanical properties of the RWM were gathered from Zhang et al. [44] and Gan et al. [45]. The RM and BM in this model have 0.4 as their Poisson’s ratio. The RM and BM also have varied Young’s moduli and damping factors along their lengths [46]. Exponential equations describing these quantities were selected to ensure the model’s results most closely resembled the experimental results used as a baseline. These equations were determined through repeat simulation using different plausible functions dependent on position along the cochlea. The material properties of the membranous labyrinth were determined in a similar manner, especially the β damping factor. This was necessary to account for the absence of anchor points, which attach the membranous labyrinth to the bony labyrinth, as the geometry of these anchor points are poorly defined in the literature.

The material properties of the cochlear implant electrode array were based on the Nucleus Straight electrode array, as the material properties of the MED-EL models are not published. A Young’s modulus of 0.4 MPa and density of 3400 kg/m^3^ were used as previously used by Lim et al. in their finite element model of residual hearing after cochlear implantation [20]. The damping factor of cochlear implant electrode arrays has not been published and was thus assumed to be that of the carrier material, silicone rubber [47].

The material properties of the endolymph and perilymph were assumed to be identical given their similar compositions. These properties were assigned as reported by Shen [48].

To the best of our knowledge, there is no published description of inner ear bone density for chinchillas. Therefore, the density of all osseous tissue was assumed to be 1200 kg/m^3^ as previously reported by Gan in her human cochlea model with further support from Wang et al.’s conclusion that chinchilla bones have a lower density than human bones [42,49]. The Young’s modulus used for osseous tissue was 14.1 GPa as in the human cochlea model reported by Wang et al. The material properties of the RWM were obtained from Gan’s model of sound transmission from the ear canal to the cochlea [42]. The elastic modulus and β damping coefficient of the OWM were identical to the RWM; however, in this model, these properties were unimportant given that the OWM was assigned a set displacement for each trial. The material properties of the cupulae were assigned according to a prior computational model of the inner ear, which studied vestibulo–cochlear interaction [41]. The material properties of the maculae were assigned according to those reported by a model that isolated the maculae of the otolith organs and separated them into two distinct layers, as was performed in this study [34].

**Table 2 bioengineering-10-00539-t002:** Mechanical Properties of the Model.

Structure	Parameter	Source
Basilar Membrane		
Density ($kg/m^{3}$)	1 × $10^{3}$	[46]
Elastic Modulus (Pa)	$7.1\times 10^{4}\times e^{-0.21x}$	
β Damping Coefficient	$2.3\times 10^{-8}\times e^{0.52x}$	
Reissner’s Membrane		
Density	1 × $10^{3}$	[46]
Elastic Modulus	$10^{4}\cdot x$	
β Damping Coefficient	$6\times 10^{-6}\times e^{0.158\cdot x}$	
Cupulae		
Density	1 × $10^{3}$	[41]
Elastic Modulus	2.8	
Maculae		
Gel Layer:		[34]
Density	1 × $10^{3}$	
Elastic Modulus	10	
Otoconial Layer:		
Density	2.71 × $10^{3}$	
Elastic Modulus	500	
Bone		
Density	1.2 × $10^{3}$	[42,49]
Elastic Modulus	13.4 × $10^{10}$	
β Damping Coefficient	0.45	
Membranous Labyrinth		
Density	1 × $10^{3}$	[41]
Elastic Modulus	1.3 × $10^{4}$	
β Damping Coefficient	0.14	
Lymphatic Fluids		
Density	1 × $10^{3}$	[41,48]
Elastic Modulus	2.6 × $10^{9}$	
β Damping Coefficient	1.5 × $10^{-4}$	
Viscosity (Pa·s)	1 × $10^{-3}$	
Speed of Sound (m/s)	1498	
Oval Window Membrane		
Density	1 × $10^{3}$	[45]
Elastic Modulus	3.5 × $10^{5}$	
β Damping Coefficient	5 × $10^{-4}$	
Round Window Membrane		
Density	1.5 × $10^{3}$	[44,45]
Elastic Modulus	3.5 × $10^{5}$	
β Damping Coefficient Cochlear Implant	5 × $10^{-4}$	
Density	3.4 × $10^{3}$	[20,47]
Elastic Modulus	4 × $10^{5}$	
β Damping Coefficient	7.7 × $10^{-2}$	

### 2.5. Boundary Conditions

The simulation was carried out in Ansys. The outer bounds of the model lie in the division between the lymphatic fluids of the inner ear and the bony labyrinth. To approximate the rigidity of the bony labyrinth, this outer surface was fixed. The fluid–solid interfaces were defined for each solid face in contact with the endolymph or perilymph. Both the endolymph and perilymph were defined as acoustic bodies to ensure the propagation of acoustic waves. A harmonic acoustic simulation was conducted to assess the BM displacement with the displacement of the stapes footplate used as the input. Experimentally determined parameters were used for the amplitude and frequency of the stapes displacement at 90 dB [42]. The BM displacement perpendicular to its surface was determined and normalized with the stapes footplate displacement for analysis. The boundary conditions of the healthy and implanted models were identical aside from additional fluid–solid interfaces being defined on the outer surfaces of the cochlear electrode.

## 3. Results

Figure 7a shows the raw, unnormalized displacements of the basilar membrane at each tested frequency. Figure 7b shows the normalized magnitude of the displacements along the cochlea. Excess noise was suppressed by applying a local filter across every 0.2 mm of the cochlea. This noise is to be expected at the overlaps of the curves between two frequencies due to their differing wavelengths [50]. The majority of noise occurs towards the end of the cochlea as acoustic waves disperse, as shown in Figure 7a. This model may generate noisier data due to the accurate triangular shape of the scala media. The magnitude of the displacement of the basilar membrane decreases as frequencies become lower. This phenomenon can be explained by the heightened stiffness of the basilar membrane towards the base and has been observed in other studies [51,52,53].

The model’s integrity was verified by comparison with the data set contained in the 1990 Greenwood study, a common source for data on the frequency and position-dependent displacement of the basilar membrane. Figure 7c shows the tuning effect of the model, gauged by the locations of maximum displacement [54]. The plot exhibits a mostly linear, downward trend in the magnitude of the displacement as the frequency decreases and is very similar in the locations of the maximum displacement for all assessed frequencies compared with the published results. This was important for the analysis of the implanted model, as the tuning effect of the cochlea is vital to the proper perception of pitch [50,55].

Figure 8 shows the data collected from the simulation with the cochlear implant. Results were collected for insertion angles between 180 and 900 degrees in increments of 180 degrees. The general trend in this model as the insertion angle increases is clear: CI surgery has the potential to have little effect on residual hearing. The magnitudes of displacement varied only slightly with all insertion angles, and the locations of the maximum displacement were almost exactly consistent with the control. Only the results for a 180-degree insertion and 900-degree insertion are shown for the sake of brevity, as all trials were very similar.

There were two major findings from these results: (1) The tuning effect of the cochlea is not significantly altered after the insertion of cochlear electrodes, representing an accurate perception of pitch; and (2) the magnitudes of the displacements in the basilar membrane are not significantly altered by the insertion of the cochlear electrodes, representing an accurate perception of volume.

## 4. Discussion

Finite element analysis of the model in this study has two major implications: (1) Cochlear implant surgery can have a minimal effect on residual hearing; and (2) the insertion angle of CIs, apart from their potential to physically damage the cochlea, has little effect on residual hearing. The literature on this topic is divided. Some studies found cochlear implant surgery to have an effect on residual hearing, such as Gan’s model on the subject [42]; others found little to no long-term effect [51]. The general consensus is that, without tip fold-over, displacement of the electrode, or some other fault, the effect of CI surgery on residual hearing is minimal [56,57]. Our results agree strongly with this conclusion. Insertion trauma is common, but it is not inevitable. While this model does represent the best-case scenario of implantation, it does not detract from the applicability to electrode design. Furthermore, as surgery techniques continue to improve and additional technology is utilized, the rate of complications is expected to decrease, and thus, the model will become more applicable.

It is desirable to create longer electrodes, as they allow the patient to sense a wider range of frequencies more effectively [8,52]. Results from this study support the theory that cochlear implants can have a minimal effect on the mechanics of the basilar membrane, even when inserted into the apex of the cochlea. These results can inform CI electrode design; longer, more slender CI electrode designs should be prioritized to preserve residual hearing function.

It was surprising that basilar membrane displacement was nearly unaffected in our model regardless of the insertion angle. Iso-Mustajärvi’s 2019 study provides one explanation [57]. He asserts that the primary contributor to the loss of residual hearing function post cochlear implantation, absent trauma, is the stiffening of the round window membrane. However, it may be that, by tuning the model for optimal results in its healthy ear setting, some sensitivity to change was lost. A stiffer model of the RWM in the implanted state and a less stiff model of the BM may provide more answers. Further study will be required to confirm or deny this possibility.

There are many future applications for this model. The addition of the vestibulocochlear nerve would allow an analysis of the electrical stimulation of the spiral ganglion with a CI electrode. Data from the electrical simulation could help further CI design by refining electrodes to reduce current spread. A vestibular implant electrode could be designed and added to examine its mechanical effect on residual hearing and balance. Vestibular implant electrodes could also be optimized to reduce current spread. This model is capable of being attached to a developed model of the chinchilla middle ear and outer ear. A combined middle ear and inner ear model could allow for an analysis of middle ear infections, such as otitis media, by modeling the middle ear cavity as being filled with fluid. The middle ear transfer function in various scenarios could be recorded to aid the clinical diagnosis of otitis media, other ear disorders, and damage to middle ear ligaments during CI surgery. Compared to simplified two-chambered or straight cochlear models, the three-chamber spiral cochlear model provides a more accurate representation of inner ear mechanics. This design introduces vital conditions, which differentiate between forward- and backward-driving conditions. These details give three-chambered models of the cochlea the capacity to exhibit a more realistic sensitivity to changing states of the inner ear. This renders them more widely applicable to further research. For example, the effect of cochlear implant surgery on residual balance could be studied using the present model and similar boundary conditions.

In the future, a series of models derived from different species will continue to be developed in our lab to enhance inner ear implantable device design and evaluation of residual hearing and balance. With the addition of the nervous system, models in a variety of species will be capable of simulating electrical stimuli. Simulations in different species will assist electrode design at various stages (e.g., initial design in rodents, further optimization in primates, and clinical trials in humans).

## Figures and Tables

**Figure 2 bioengineering-10-00539-f002:**
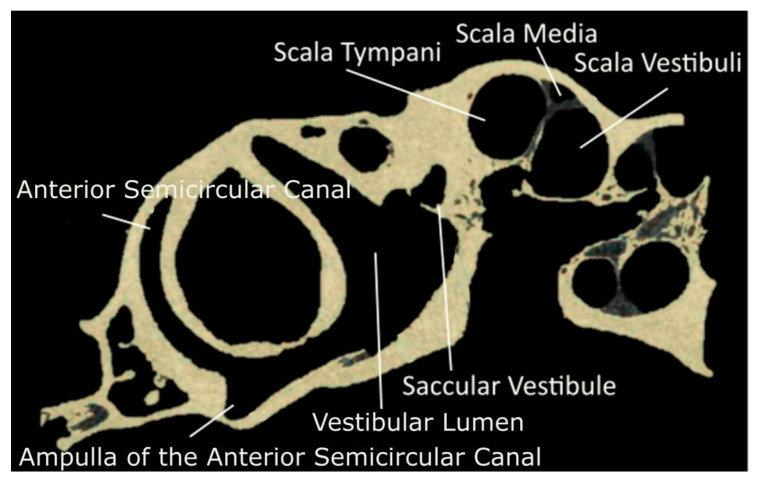
Segmented µCT scan in the sagittal plane of the chinchilla subject with key structures labeled. Refer to Table 1 for symbol definitions.

**Figure 3 bioengineering-10-00539-f003:**
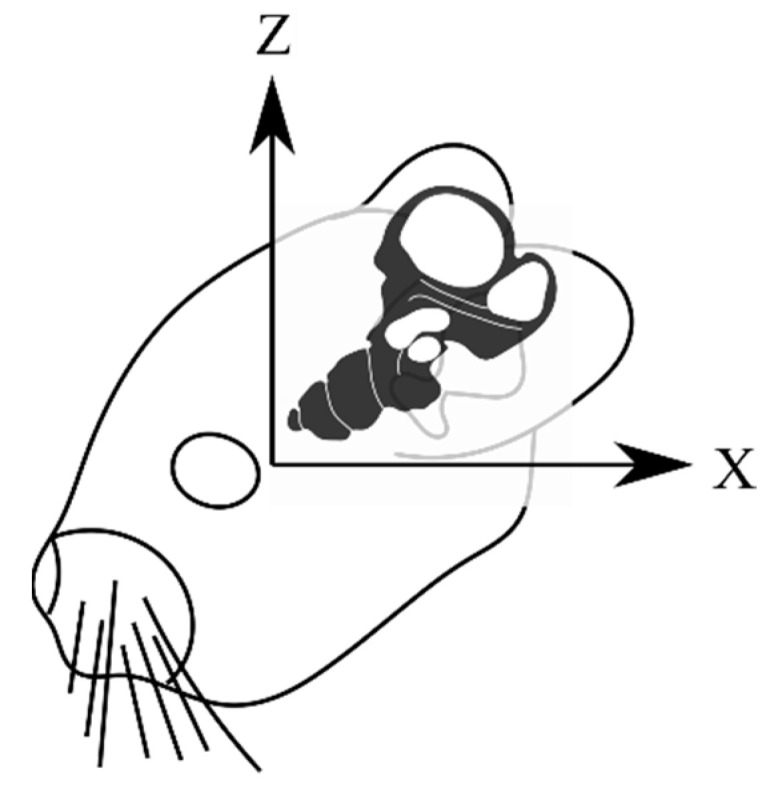
The coordinate system used in all imaging for this paper. The x and z axes are held in the sagittal plane as if viewed from the subject’s left side.

**Figure 4 bioengineering-10-00539-f004:**
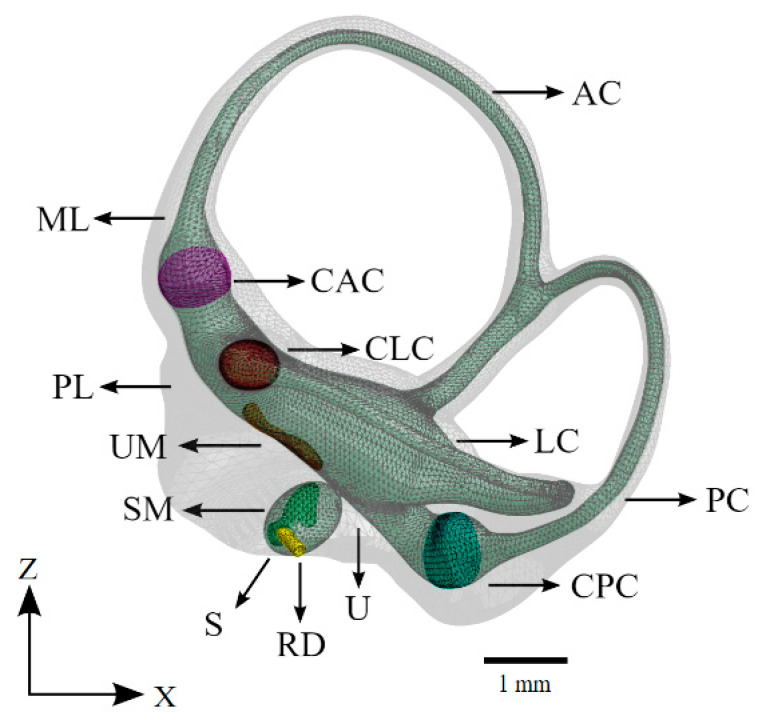
The vestibular system of the computational model. The saccule, utricle, and semicircular canals appear as a continuous volume of lymphatic fluid (green). The sensory organs of the vestibular system are also shown. Refer to Table 1 for symbol definitions.

**Figure 5 bioengineering-10-00539-f005:**
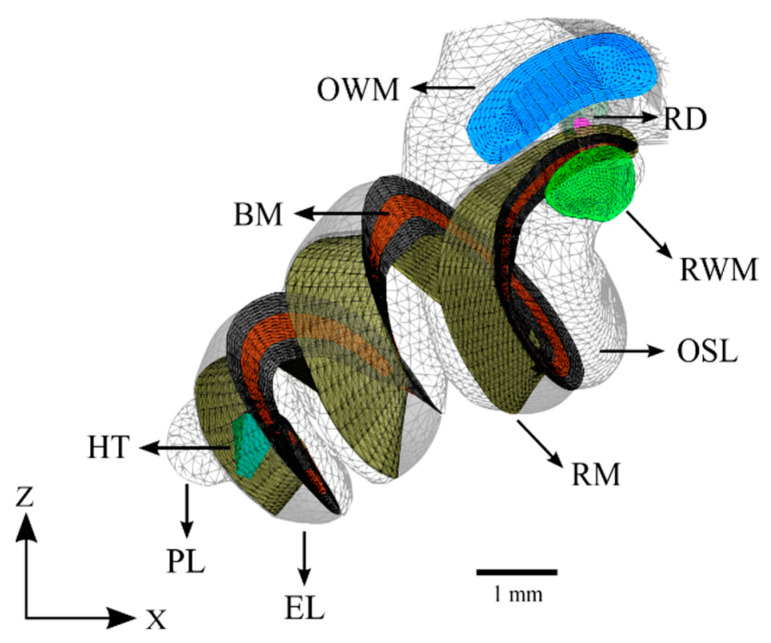
The cochlea of the computational model without the cochlear implant. The design of the basilar membrane (red) is apparent as a ribbon with varying width and thickness, attached on both sides to bony supports (grey). The distal side of the basilar membrane is visible as the attachment point for the end of the Reissner’s membrane. Refer to Table 1 for symbol definitions.

**Figure 6 bioengineering-10-00539-f006:**
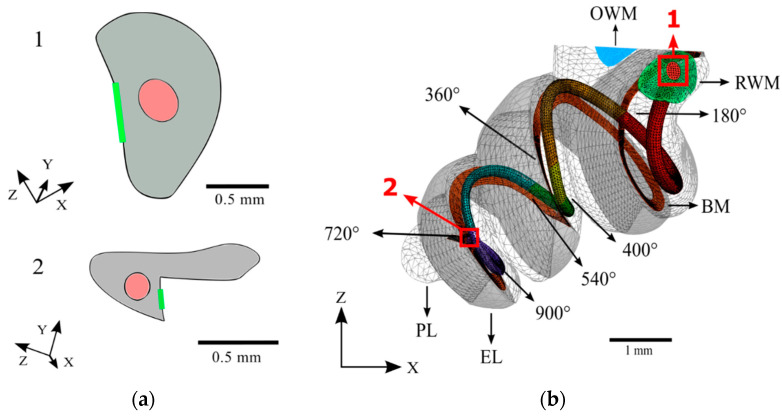
The full meshed model of the cochlea is presented with the length of the cochlear implant electrode inserted. (**a**) Cross-sections of the base (1) and apex (2) ends of the cochlear implant electrode; (**b**) The path of the cochlear implant electrode through the scala tympani of the meshed model.

**Figure 7 bioengineering-10-00539-f007:**
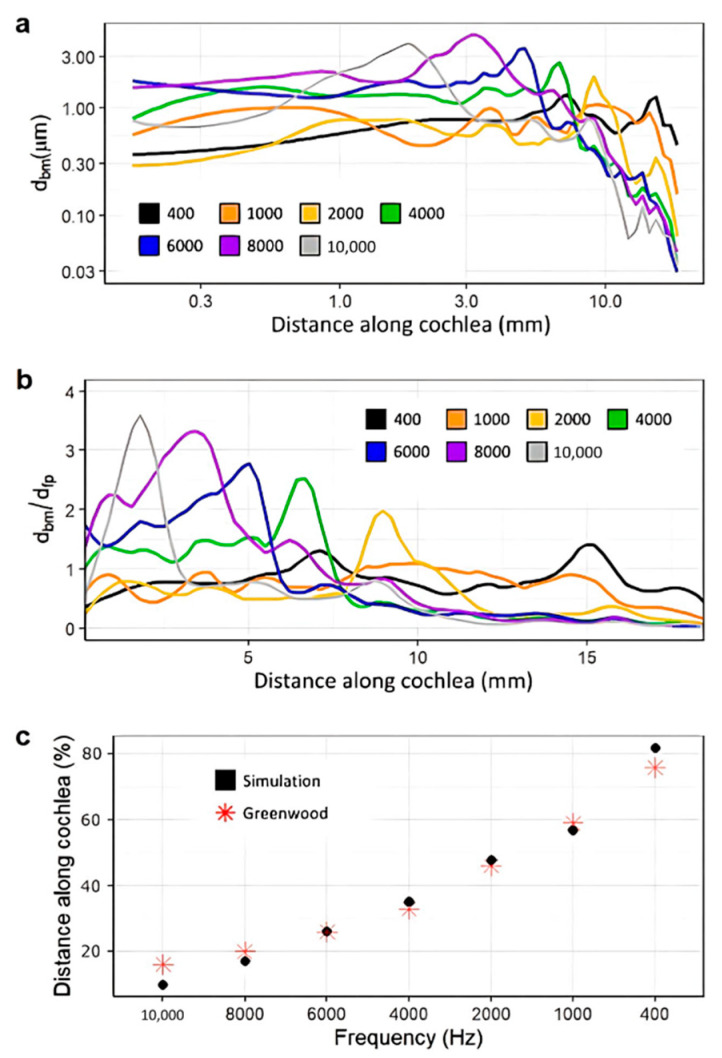
Displacement of the basilar membrane from the base to the apex of the cochlea before insertion of the CI (400 Hz: black, 1000 Hz: orange, 2000 Hz: yellow, 4000 Hz: green, 6000 Hz: blue, 8000 Hz: purple, 10,000 Hz: grey). Model input was the experimentally determined frequency dependent displacement of the stapes at 90 dB [42]. (**a**,**b**) show unnormalized displacement of the basilar membrane and basilar membrane displacement normalized with that of the stapes footplate, respectively. (**c**) shows the location of the maximum displacement for the model (black) at each frequency compared to an experimentally obtained benchmark [54].

**Figure 8 bioengineering-10-00539-f008:**
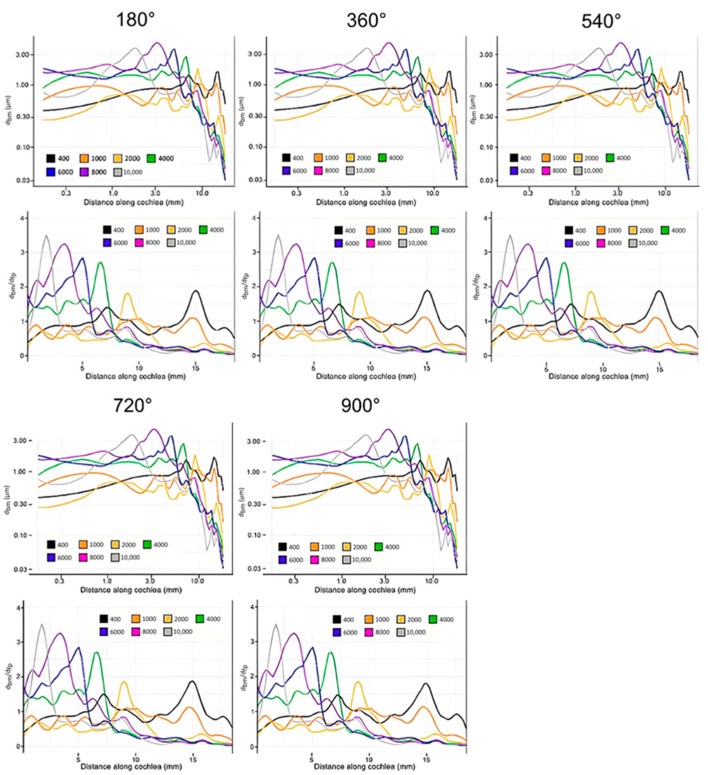
Displacement of the basilar membrane from the base to the apex of the cochlea at each evaluated insertion angle of the cochlear electrode (400 Hz: black, 1000 Hz: orange, 2000 Hz: yellow, 4000 Hz: green, 6000 Hz: blue, 8000 Hz: purple, 10,000 Hz: grey). Model input was the experimentally determined frequency dependent displacement of the stapes at 90 dB [42]. The upper figure in each set and the lower set show unnormalized displacement of the basilar membrane and basilar membrane displacement normalized with that of the stapes footplate, respectively.

**Table 1 bioengineering-10-00539-t001:** List of Abbreviations Related to Cochlear Implantation.

EL	Endolymph	UM	Utricular Macula
PL	Perilymph	SM	Saccular Macula
BM	Basilar Membrane	SCC	Semicircular Canal
RM	Reissner’s Membrane	AC	Anterior Semicircular Canal
OSL	Osseous Spiral Lamina	PC	Posterior Semicircular Canal
HT	Helicotrema	LC	Lateral Semicircular Canal
RD	Reuniting Duct	CAC	Cupula of the AC
ML	Membranous Labyrinth	CPC	Cupula of the PC
RWM	Round Window Membrane	CLC	Cupula of the LC
OWM	Oval Window Membrane	CI	Cochlear Implant
U	Utricle	CIE	Cochlear Implant Electrode
S	Saccule	CIS	Cochlear Implant Surgery
SCC	Semicircular Canal		

## Data Availability

The data presented in this study are available on request from the corresponding author. The data are not publicly available due to the policy of the University of Oklahoma.

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
