# Peer review of "Finite Element Modeling of Residual Hearing after Cochlear Implant Surgery in Chinchillas"

_bioengineering, 2023, doi:10.3390/bioengineering10050539_

Round 1
Reviewer 1 Report
The recent works on 3D modelling of cochlear implantation should be referenced and discussed, for example:
Ren, LJ., Yu, Y., Zhang, YH. et al. Three-dimensional finite element analysis on cochlear implantation electrode insertion. Biomech Model Mechanobiol (2022). https://doi.org/10.1007/s10237-022-01657-3
Potrusil et al., Finite element analysis and three-dimensional reconstruction of tonotopically aligned human auditory fiber pathways: A computational environment for modeling electrical stimulation by a cochlear implant based on micro-CT, https://doi.org/10.1016/j.heares.2020.108001
The description of FEM model should be extended by providing more information about elements "..Ansys Solid185 element type while the fluids were modelled using the Ansys Fluid40 element type..." For non-Ansys users the element type number do not give any information.
The paper is well written and can be published after minor modifications.
Author Response
Hello, thank you for your helpful comments and the sources you provided. We have updated our manuscript to address your concerns in the following ways:
The recent works on 3D modelling of cochlear implantation should be referenced and discussed, for example:
Ren, LJ., Yu, Y., Zhang, YH. et al. Three-dimensional finite element analysis on cochlear implantation electrode insertion. Biomech Model Mechanobiol (2022). https://doi.org/10.1007/s10237-022-01657-3
Potrusil et al., Finite element analysis and three-dimensional reconstruction of tonotopically aligned human auditory fiber pathways: A computational environment for modeling electrical stimulation by a cochlear implant based on micro-CT, https://doi.org/10.1016/j.heares.2020.108001
- We have updated the prior FE models section to include the Ren paper you suggested, as well as one other paper on the subject. The added content should be below. We opted not to include the article on electrical stimulation due to our focus on mechanical residual hearing, however I did include it in my thesis, thank you for the recommendation. Our specific changes are below:
- “One such model found that material, geometric design, insertion speed, and friction coefficients were the greatest factors influencing residual hearing preservation [52].”
- “This model provided a good first step towards further FE analysis of residual hearing after CI surgery, though did not necessarily agree with the results of other models where residual hearing was found to be less effected by the simple presence of a CI electrode and more effected by the trauma caused during insertion [51, 52]”
The description of FEM model should be extended by providing more information about elements "..Ansys Solid185 element type while the fluids were modeled using the Ansys Fluid40 element type..." For non-Ansys users the element type number do not give any information.
- Thank you for pointing this out. We have updated the meshing section to include some basic information on these two element types, including their number of nodes, degrees of freedom, and common applications. The added content is below.
- “All elements were tetrehedral. Both SOLID185 and FLUID30 tetrahedral elements have 8 nodes, each with 3 degrees of freedom. SOLID185 is a very commonly applied element type in ANSYS, often in structural analysis and in fluid-structure interactions. FLUID30 is accepted as a standard element type for fluids in simulation of fluid-structure interaction.”
Thank you very much,
Nick Castle
Reviewer 2 Report
The paper is well written and can be published after some modifications.
Please find below some additional comments, as you suggest.
1. What is the main question addressed by the research?
Main question addressed by the present research is to present a finite element (FE) model of the chinchilla inner ear for studying the interrelationship between the mechanical function and the insertion angle of a CI electrode.
2. Do you consider the topic original or relevant in the field? Does it
address a specific gap in the field?
The topic is relevant, due to the fact that in the case of cohlear implant surgery the effects of a successful scala tympani insertion on the mechanics of hearing are not yet fully understood.
3. What does it add to the subject area compared with other published
material?
This study focuses on the effect of cochlear electrode insertion depth on the residual mechanical function of the cochlea in a chinchilla computational model. The unimplanted model is demonstrated here first and compared to expected response curves to demonstrate its initial validity. The analysis will then focus on the effects of cochlear implantation on residual hearing.
4. What specific improvements should the authors consider regarding the
methodology? What further controls should be considered?
The methodology is all right. The chinchilla is commonly used as an analog for the human in hearing and balance studies due to its similar number of turns in the cochlea, structure of semicircular canals, singular primary crista, and hearing range.
The authors could improve the part dedicated the material and their properties that is used during FEM analysis. More details must be added, in my opinion. If the data shown in table 2 are collected from literature, add a new column with references (Table 2. Mechanical Properties of the Model.).
E.g.
- The damping factor of cochlear implant electrode arrays has not been published and was thus assumed to be that of the carrier material, silicone rubber.
- Material properties of the endolymph and perilymph were assumed to be identical given their similar compositions.
- Therefore, the density of all osseous tissue was assumed to be 1200 kg/m3 as previously done by Gan in her human cochlea model with further support from Wang et al.’s conclusion that chinchilla bones have a lower density than human bones.
5. Are the conclusions consistent with the evidence and arguments presented and do they address the main question posed?
The conclusions consistent with the evidence and arguments presented.
6. Are the references appropriate?
The references are appropriate.
7. Please include any additional comments on the tables and figures.
- If the data shown in table 2 are collected from literature, add a new column with references (Table 2. Mechanical Properties of the Model.).
- Modify the description for Table 1. I suggest to use “Table 1. List of Abbreviations related to the cohlear implantation” due to the fact that other abbreviations are used and not described (e.g. MRI, CT).
- regarding the figure 2, I suggest to not use abbreviation (AC could be mentioned as “Anterior Semicircular Canal” on the figure: also, a mistake appear related to the “Ampulla of SC”; SC is not defined in table 1).
- regarding the figure 3 “Figure 3. The coordinate system used in all imaging for this paper. The x and z axes are held in the sagittal plane as if viewed from the subjects left side.”, in this figure appear x and y axes. Please, modify the figure and legend according to the other figure (fig. 4 and 5 shown x, z axes; figure 6 shown x.y.z axes.
Author Response
- The authors could improve the part dedicated to the material and their properties that is used during FEM analysis. More details must be added, in my opinion. If the data shown in table 2 are collected from literature, add a new column with references (Table 2. Mechanical Properties of the Model
- Thank you for your input. We have updated Table 2 with a column for the source of each material property used in our model. We have also gone into more depth in the material property section, discussing the material properties of the round window membrane, oval window membrane, maculae, and cupulae. The added content is below.
- “Material properties of the RWM were obtained from Gan’s model of sound transmission from the ear canal to the cochlea [34]. The elastic modulus and β damping coefficient of the OWM were identical to the RWM, however in this model these properties were unimportant given the OWM was assigned a set displacement for each trial. Material properties of the cupulae were assigned according to a similar computational model of the inner ear which studied vestibulo-cochlear interaction [34]. Material properties of the maculae were assigned according to those reported by a model which isolated the maculae of the otolith organs and separated them into two distinct layers, as was done in this study [50].”
- “Material properties of the membranous labyrinth were determined in a similar manner, especially the β damping factor. This was necessary to account for the absence of anchor points which attach the membranous labyrinth to the bony labyrinth, as the geometry of these anchor points are poorly defined in literature.”
- Modify the description for Table 1. I suggest to use “Table 1. List of Abbreviations related to the cochlear implantation” due to the fact that other abbreviations are used and not described (e.g. MRI, CT).
- This has been corrected, Table 1 is now titled “List of Abbreviations Related to Cochlear Implantation”
- regarding the figure 2, I suggest to not use abbreviation (AC could be mentioned as “Anterior Semicircular Canal” on the figure: also, a mistake appear related to the “Ampulla of SC”; SC is not defined in table 1).
- Thank you for your suggestion. We have modified the figure so that it does not use abbreviations.
- regarding the figure 3 “Figure 3. The coordinate system used in all imaging for this paper. The x and z axes are held in the sagittal plane as if viewed from the subjects left side.”, in this figure appear x and y axes. Please, modify the figure and legend according to the other figure (fig. 4 and 5 shown x, z axes; figure 6 shown x.y.z axes.
- We have modified the figure to show the correct axes, thank you for pointing out that mistake.
Thank you very much and please let us know if you have any further concerns,
Nick Castle
Reviewer 3 Report
This paper presents a finite element (FE) model of the chinchilla inner ear for studying the interrelationship between the mechanical function and the insertion angle of a cochlear implant electrode. This model includes a three-chambered cochlea and full vestibular system, accomplished using µ-MRI and µ-CT scanning technology. The manuscript is well-written, the model characterization is thorough, and the figures are appealing. However, the manuscript has some issues, which need to be fully addressed, before it can be considered for publication in Bioengineering.
1. Although the author claimed in the Discussion part (page 11, line 294-313) that this model has many applications in the future, it did not show it in the manuscript. I suggest that the authors add application experiments to support the author's point.
2. The authors need to explain the advantages of their model over other methods in the manuscript. A table or radar charts was recommended to be added in the manuscript.
The amended paper can be considered for publication.
Author Response
- Although the author claimed in the Discussion part (page 11, line 294-313) that this model has many applications in the future, it did not show it in the manuscript. I suggest that the authors add application experiments to support the author's point.
- Thank you for your input. We have amended the manuscript and believe that the following information should be sufficient to demonstrate the future applications of this model. We have listed the following applications:
- Modeling of residual mechanical hearing after vestibular implantation
- Modeling of electrical stimulation by cochlear implant electrodes to reduce current spread
- Modeling of electrical stimulation by vestibular implant electrodes to reduce current spread
- The addition of the middle ear to simulate the effect of otitis media and trauma during cochlear implant surgery on the middle ear transfer function
- The authors need to explain the advantages of their model over other methods in the manuscript. A table or radar charts was recommended to be added in the manuscript.
- Because of the importance of this topic, we have opted to add a new section, 1.4. Advantages over In Vivo testing. We hope that this sufficiently answers your question as we struggled to format this volume of information into an easy to understand table or to quantitatively measure the advantages of the FE method over In Vivo testing. The added section is below. Thank you.
- 1.4. Advantages of the FE Method over In Vivo Testing
- The use of laboratory animals is a necessary part of medical research, as it enables scientists to explore new treatments before progressing to human trials. However, animal testing is a complex issue that raises many ethical and logistical concerns, particularly regarding the welfare and cost of laboratory animals. Animal testing must be carried out in accordance with strict ethical guidelines to ensure that any suffering is minimized [53]. In medical research it is often necessary to purchase many expensive research-grade animals, making their use particularly expensive, especially when considering long-term management of an animal facility [54]. Furthermore, the quality of laboratory animals can be compromised by unethical practices, making studies less productive and reproducible. Finite element modeling is a solution to both the monetary and ethical problems involved in animal research [55, 56]. FE modeling is cost-effective, ethical, reproducible, and safe. Models can be precisely manipulated at will in a relatively short time frame to account for a variety of different variables and conditions, some of which may not be foreseen prior to beginning modeling. Simulations can be run as many times as researchers desire with little-to-no variation in the model’s geometry between iterations, something impossible when using multiple animals in a study [57]. In animal testing this kind of iterative process is also be quite expensive, involving the purchase of many animals. The FE method can be applied without any harm to the animal subjects as medical imaging of delicate structures can be obtained non-invasively. Imaging can be shared among institutions, further reducing the number of animals needed for FE modeling. While not a replacement for animal testing, it is clear that in early stages of research the FE method should be explored prior to In Vivo testing on animal or human subjects.
Thank you very much for your helpful comments and please let us know if you have further concerns you would like addressed,
Nick Castle